# Transferability of genetic loci and polygenic scores for cardiometabolic traits in British Pakistani and Bangladeshi individuals

Qin Qin Huang [1,18], Neneh Sallah[2,3,18], Diana Dunca[2,3], Bhavi Trivedi[4], Karen A. Hunt[4], Sam Hodgson [5], Samuel A. Lambert [6,7,8], Elena Arciero[1], John Wright[9], Chris Griffiths [10], Richard C. Trembath [11], Harry Hemingway [2,12,13], Michael Inouye [6,7,8,14,15], Sarah Finer [4], David A. van Heel [4], R. Thomas Lumbers [2,13,16,19], Hilary C. Martin [1,19] & Karoline Kuchenbaecker [3,17,19] ✉

Individuals with South Asian ancestry have a higher risk of heart disease than other groups but have been largely excluded from genetic research. Using data from 22,000 British Pakistani and Bangladeshi individuals with linked electronic health records from the Genes & Health cohort, we conducted genome-wide association studies of coronary artery disease and its key risk factors. Using power-adjusted transferability ratios, we found evidence for transferability for the majority of cardiometabolic loci powered to replicate. The performance of polygenic scores was high for lipids and blood pressure, but lower for BMI and coronary artery disease. Adding a polygenic score for coronary artery disease to clinical risk factors showed significant improvement in reclassification. In Mendelian randomisation using transferable loci as instruments, our findings were consistent with results in European-ancestry individuals. Taken together, trait-specific transferability of trait loci between populations is an important consideration with implications for risk prediction and causal inference.

Individuals with South Asian ancestry account for more than a fifth of the global population and experience a higher risk of coronary artery disease (CAD) than other ancestries. For example, British South Asians have three- to four-fold higher CAD risk than White British people[1].

[1]Department of Human Genetics, Wellcome Sanger Institute, Cambridge, UK. [2]Institute of Health Informatics, University College London, London, UK. [3]UCL Genetics Institute, University College London, London, UK. [4]Blizard Institute, Barts and the London School of Medicine and Dentistry, Queen Mary University of London, London, UK. [5]Primary Care Research Centre, University of Southampton, Southampton, UK. [6]Cambridge Baker Systems Genomics Initiative, Department of Public Health and Primary Care, University of Cambridge, Cambridge, UK. [7]British Heart Foundation Cardiovascular Epidemiology Unit, Department of Public Health and Primary Care, University of Cambridge, Cambridge, UK. [8]Health Data Research UK Cambridge, Wellcome Genome Campus and University of Cambridge, Cambridge, UK. [9]Bradford Institute for Health Research, Bradford Teaching Hospitals National Health Service (NHS) Foundation Trust, Bradford, UK. [10]Institute of Population Health Sciences, Barts and the London School of Medicine and Dentistry, Queen Mary University of London, London, UK. [11]Department of Medical and Molecular Genetics, King's College London, London, UK. [12]Health Data Research UK, University College London, London, UK. [13]University College London Hospitals Biomedical Research Centre (UCLH BRC), London, UK. [14]British Heart Foundation Cambridge Centre of Research Excellence, Department of Clinical Medicine, University of Cambridge, Cambridge, UK. [15]Cambridge Baker Systems Genomics Initiative, Baker Heart and Diabetes Institute, Melbourne, VIC, Australia. [16]British Heart Foundation Research Accelerator, University College London, London, UK. [17]Division of Psychiatry, University College London, London, UK. [18]These authors contributed equally: Qin Qin Huang, Neneh Sallah. [19]These authors jointly supervised this work: R. Thomas Lumbers, Hilary C. Martin, Karoline Kuchenbaecker. ✉e-mail: k.kuchenbaecker@ucl.ac.uk

Understanding the determinants of excess CAD burden in South Asian populations and improving prediction to enable preventive interventions to represent important public health priorities.

Common genetic variation is an important determinant of CAD and of upstream risk factors, such as blood pressure, lipids, and body mass index (BMI). The genetic component of disease risk can be harnessed to identify underlying disease genes and pathways, to estimate the unconfounded effects of risk factors by Mendelian randomisation, and to improve risk prediction through the application of polygenic scores (PGS). However, the genetic basis of CAD risk is not well characterised in South Asian ancestry populations because genome-wide association studies (GWAS) have been mostly limited to European ancestry populations[2].

Fundamental questions remain about the extent to which the genetic determinants of cardiometabolic traits are shared by European and South Asian ancestry populations. These have important implications for translational applications of genetic data, such as causal inference with Mendelian randomisation which could prioritise different prevention strategies or drug targets between ancestries, and clinical risk prediction. Whilst the predictive performance of PGSs derived from European ancestry populations in other ancestry groups decreases with genetic distance[3–6], the extent to which this attenuation is due to genetic drift (differences in linkage disequilibrium and allele frequency[7]) versus heterogeneity of causal genetic effects remains unclear.

Furthermore, most previous large-scale studies assessed genetic risk prediction using data from a research setting. These findings may not generalise well to a real-world clinical setting. Firstly, clinical risk factors may be measured less comprehensively than in a research setting, affecting the performance of integrated risk models combining these factors with PGSs[8–10]. Secondly, there is evidence that the performance of PGSs may be modified by factors, such as educational attainment and socioeconomic status for which research studies are often not representative[11,12]. The robustness of PGSs applied to South Asian-ancestry individuals in a real-world healthcare system is largely unknown.

Here, we perform a comparative analysis of the genetics of CAD and upstream cardiometabolic traits in European and South Asian ancestry populations, using data from the Genes & Health (G&H) cohort[13]. G&H is a community-based cohort of British Pakistani and Bangladeshi individuals with linked electronic health record data ($N = 22,490$ individuals). This unique cohort represents an understudied and clinically vulnerable population with high levels of socioeconomic deprivation, and this is, to our knowledge, the first major genetic study focused on it. We apply new approaches to the transferability of genomic risk loci across populations, perform ancestry-specific and trans-ancestry Mendelian randomisation analysis, investigate the transportability of PGSs for CAD and its risk factors, and estimate the incremental improvement in CAD prediction when incorporating the CAD PGS into clinical risk tools.

## Results

We conducted GWAS of CAD and key cardiometabolic traits in the G&H cohort which was the primary data resource for this study. In G&H, 4.9% ($N = 1110$) of the individuals had coronary artery disease (CAD), with the age of onset ranging from 17 to 97 years old (median 55). A quarter of the G&H participants were on active statin prescriptions, 23% on BP medications, 29% had high TC levels (>5 mmol/L), and 30% had high LDL-C levels (>3 mmol/L; Supplementary Data 6).

We used publicly available GWAS summary statistics derived from predominantly European ancestry individuals to compare the genetic architecture and assess whether reported GWAS loci are transferable to G&H (Fig. 1). We evaluated the performance of European ancestry-derived PGSs in G&H and compared it to performance in European ancestry samples from eMERGE. Finally, we used Mendelian randomisation analysis to test the causal relationship between the cardiometabolic traits and CAD by comparing genetic instruments based on the GWAS data generated from European ancestry and from British South Asian ancestry individuals. The datasets that were used in each analysis are described in Supplementary Data 4.

### Shared genetic architecture of cardiometabolic traits

We compared the genetic architecture of CAD and upstream risk factors, namely HDL-C, LDL-C, triglycerides (TG), total cholesterol (TC), systolic and diastolic blood pressure (SBP & DBP), between British Pakistanis and Bangladeshis from G&H, and European ancestry individuals from the electronic health record-based eMERGE cohort, since phenotypes had been ascertained in a similar way. All traits were found to have significant SNP heritability ($h^2 = 0.04–0.20$) in G&H, with estimates similar to those in eMERGE (Supplementary Data 7, Fig. 2a),

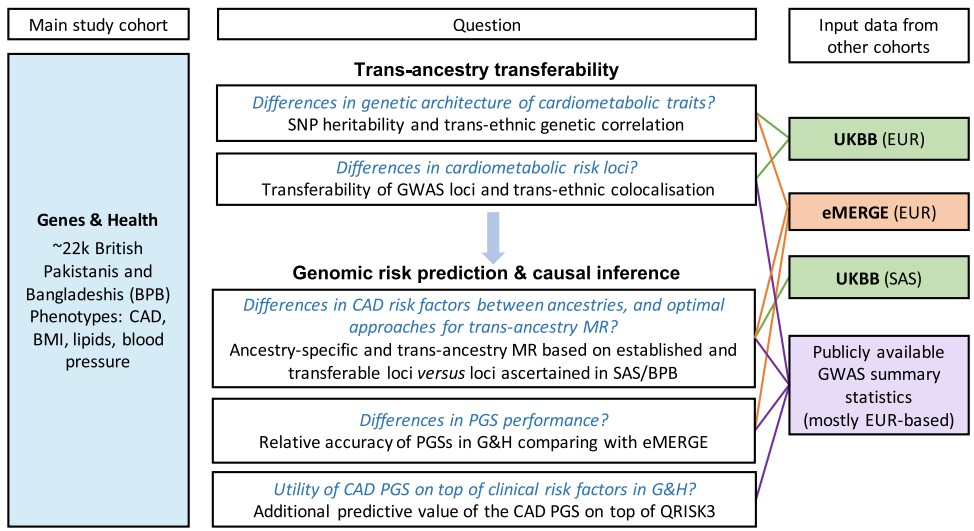

**Fig. 1 | Summary of study design, research questions and analyses conducted.** The coloured boxes indicate input data. Within the white boxes, the black text indicates the analyses we used to address the questions in blue. BPB British Pakistanis and Bangladeshi ancestry, EUR European ancestry, SAS South Asian ancestry, CAD coronary artery disease, BMI body mass index, SNP single nucleotide polymorphism, GWAS genome-wide association study, MR Mendelian randomisation, PGS polygenic score, UKBB UK Biobank. Datasets and discovery GWAS that were used in each analysis are provided in Supplementary Data 4.

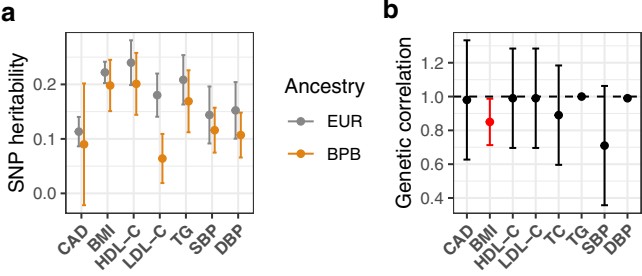

**Fig. 2 | SNP heritability and trans-ancestry genetic correlations for cardiometabolic traits. a** SNP heritability was estimated using GCTA in G&H (orange) and eMERGE (grey) for cardiometabolic traits, namely coronary artery disease (CAD; *n* = 17,348 and 32,816 unrelated samples from G&H and eMERGE, respectively), body-mass index (BMI; *n* = 13,926 and 37,160), high-density lipoprotein cholesterol (HDL-C; *n* = 11,316 and 16,049), low-density lipoprotein cholesterol (LDL-C; *n* = 12,856 and 15,856), triglycerides (TG; *n* = 11,125 and 14,384), systolic blood pressure (SBP), and diastolic blood pressure (DBP; *n* = 15,908 and 11,864 for blood pressure). Medication data are not available in eMERGE so the highest measurements for LDL-C, SBP, and DBP are used. Error bars represent 95% confidence intervals in both plots. **b** Genetic correlations were estimated using Popcorn based on GWAS summary statistics generated from G&H and European-ancestry individuals from UK Biobank. Red indicates that the genetic correlation is nominally significantly lower than 1 (*p*-value = 0.02 for BMI; two-sided and not adjusted for multiple comparisons). Medication-adjusted lipid and blood pressure levels are used. For $r_g$ estimates of 1 (TG and DBP), the method cannot derive confidence intervals. Sample sizes of GWAS for cardiometabolic traits in G&H are in Table 1.

except for LDL-C and blood pressure which had significantly lower values in G&H than eMERGE (e.g. for LDL-C, $h^2$ was 0.18 [95% CI: 0.14–0.22] in eMERGE and 0.06 [95% CI: 0.02–0.11] in G&H; z test one-sided $p = 7.3 \times 10^{-5}$). We observed high genetic correlations between G&H and European ancestry samples from UKBB for all traits, with the lowest value seen for SBP ($r_g = 0.71$ [95% CI: 0.36–1.06], $p = 0.09$; Fig. 2b). The only trait for which the genetic correlation differed nominally significantly from one was BMI ($r_g = 0.85$ [95% CI: 0.71–0.99], $p = 0.02$, not adjusted for multiple comparisons).

## High transferability of cardiometabolic loci

We assessed whether published trait-associated genomic loci identified in predominantly European ancestry populations were shared by the British Pakistani and Bangladeshi populations represented by G&H. To account for differences in LD patterns, our assessment of transferability was based on the credible sets of variants per locus, likely to contain the causal variant, rather than the sentinel variants alone. Low numbers of transferable loci may be due to limited statistical power

rather than a lack of causal variant sharing. Therefore, we compared the number of observed transferable loci with the number expected given the sample size and allele frequency in G&H if all causal variants were shared. The number of expected transferable loci varied widely between traits (e.g. we expected to be able to detect significant associations for 56% of HDL-C loci but only for 18% of SBP loci), highlighting the importance of accounting for power when assessing transferability. We report the observed number divided by the expected number of loci and call this new approach the power-adjusted transferability (PAT) ratio. Across most traits examined, the observed number of transferable loci closely matched the loci we expected (Table 1 and Supplementary Data 8). For example, for BMI we expected to be able to find evidence for transferability for 20% of loci and we did indeed observe transferability for 21% of loci, yielding a PAT ratio of 1.05. The PAT ratio for CAD was only 0.62, with the number of observed transferable loci (13%) lower than the expected number (21%), although this difference was only marginally significant (binomial *p*-value = 0.05; one-sided and not adjusted for the multiple comparisons). To explore whether this was likely to be due to ancestry differences or other factors, we also calculated the PAT ratio in eMERGE, and observed a similarly low PAT ratio for CAD (0.69, binomial *p*-value = $6 \times 10^{-4}$) (Table 1 and Supplementary Data 9).

We also assessed whether there were any specific loci that were not transferable despite being well powered to observe an association (power > 80%). Out of a total of 184 well-powered loci tested across all traits, only nine were non-transferable; that is, no variant in the credible set was significant at $p < 0.05$ and no variant within 50 kb of the locus was significant at $p < 1 \times 10^{-3}$ (Fig. S4). These nine loci were all associated with lipid traits: *EVI5*, *NBEAL1*, *GPAM*, *CETP*, *STAB1*, *TTC39B*, *SH2B3*, *ACP2,* and *NECAP2* (Supplementary Data 10). Of these loci, *CETP*, which has been reported to be associated with both HDL-C and LDL-C levels in European ancestry samples, was strongly associated with HDL-C in G&H ($p = 7.08 \times 10^{-56}$), but not with LDL-C levels ($p = 0.23$) (Fig. S5) despite having >80% power for replication.

Even when there are associations in the same region in two ancestry groups, it is possible that they are driven by different causal variants, as previously seen[14]. To assess the extent of sharing of causal variants between ancestries at previously reported loci with evidence of transferability, we applied trans-ancestry colocalisation for G&H with UKBB European ancestry samples as the reference. Colocalisation methods can estimate the likelihood of causal variant sharing without the need to identify the specific causal variant. We found evidence for the most extensive sharing of causal variants for transferable lipid loci: total cholesterol (61% of loci had significant colocalisation), followed by TG (56%), HDL-C (48%), and LDL-C (47%) (Table 1). For BMI we found evidence for sharing of causal variants for only 26% of transferable loci

**Table 1 | Transferability of loci for cardiometabolic phenotypes from European ancestry (EUR) discovery GWAS to British Pakistani and Bangladeshi individuals**

| Trait | No. of samples (cases:controls) | Loci associated in EUR | Observed transferable loci (%) | Expected transferable loci in % | PAT ratio (*p*-value) | Shared causal variant/loci assessed (%) | PAT ratio in eMERGE (*p*-value) |
|---|---|---|---|---|---|---|---|
| BMI | 16,890 | 662 | 140 (21%) | 20 | 1.05 (0.79) | 15/58 (26%) | 0.91 (0.05) |
| LDL-C | 12,746 | 82 | 51 (62%) | 50 | 1.24 (0.99) | 15/32 (47%) | 0.60 (1.6 × 10⁻⁵)ᵃ |
| HDL-C | 14,944 | 103 | 66 (64%) | 56 | 1.14 (0.96) | 14/29 (48%) | 0.91 (0.20) |
| TC | 15,641 | 107 | 61 (57%) | 49 | 1.16 (0.96) | 23/38 (61%) | – |
| TG | 13,037 | 95 | 47 (49%) | 47 | 1.04 (0.72) | 14/25 (56%) | 0.96 (0.35) |
| DBP | 18,536 | 175 | 36 (21%) | 23 | 0.91 (0.26) | NaN | 0.76 (0.07)ᵃ |
| SBP | 18,536 | 171 | 30 (18%) | 22 | 0.82 (0.12) | NaN | 0.77 (0.12)ᵃ |
| CAD | 22,008 (1110:20898) | 71 | 9 (13%) | 21 | 0.62 (0.05) | NaN | 0.69 (6 × 10⁻⁴)ᵃ |

Transferability was defined as a significant association of a variant in the credible set at a locus. The power-adjusted transferability (PAT) ratio is calculated as dividing the observed number of transferable loci over the expected number. One-sided *p*-values were calculated using binomial tests and were not adjusted for multiple comparisons. For transferable loci with good genotyping coverage trans-ancestry colocalisation (TAColoc) was used to evaluate whether the associations are driven by the same causal variant in both populations.
ᵃIn eMERGE, medication data were not available thus we used the highest measurements for LDL-C, SBP, and DBP. CAD was defined based on ICD10 codes only (Supplementary Methods). Total cholesterol levels were not available.

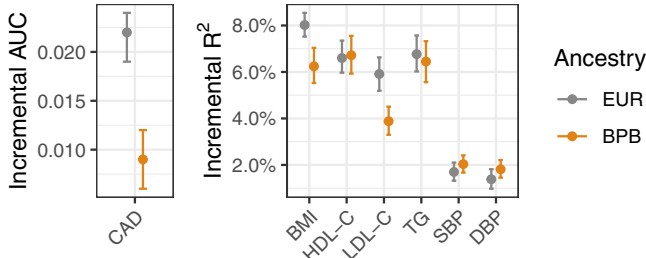

**Fig. 3 | Comparison of the predictive accuracy of polygenic scores in people of British Pakistani and Bangladeshi versus European ancestry.** Incremental AUC is shown for coronary artery disease (CAD; $n = 17,348$ and 32,816 unrelated samples from G&H and eMERGE, respectively) and Incremental $R^2$ is shown for its continuous risk factors, namely body-mass index (BMI; $n = 13,926$ and 37,160), high-density lipoprotein cholesterol (HDL-C; $n = 11,316$ and 16,049), low-density lipoprotein cholesterol (LDL-C; $n = 12,856$ and 15,856), triglycerides (TG; $n = 11,125$ and 14,384), systolic blood pressure (SBP), and diastolic blood pressure (DBP; $n = 15,908$ and 11,864 for blood pressure). Grey indicates European-ancestry (EUR) individuals from eMERGE and orange British Pakistani and Bangladeshi (BPB) individuals from G&H. Error bars represent 95% confidence intervals estimated by bootstrap resampling of samples. The highest measurements for LDL-C, SBP, and DBP are compared between eMERGE and G&H.

assessed (Table 1 and Supplementary Data 11). Causal variants in major lipid loci such as *PCSK9* were among variants that were consistently not shared ($p_{JLIM} > 0.05$) between the two populations (Fig. S6 and Supplementary Data 11).

## Variable performances of polygenic scores

To assess the performance of PGSs for cardiometabolic traits derived from European ancestry populations in British Pakistani and Bangladeshi individuals, we compared predictive performance in G&H to that in European ancestry individuals from eMERGE. We quantified predictive accuracy using the "incremental AUC" statistic for CAD and the "incremental $R^2$" statistic for continuous risk factor traits; these are the gain in AUC or $R^2$ when adding the PGS to the regression of phenotype on the baseline covariates (sex, age, and genetic PCs).

We first evaluated previously published PGSs from the PGS Catalog (Supplementary Data 5). The CAD PGSs that proved to have the best performance in G&H and eMERGE were two different scores optimised in South Asian[15] and European ancestry samples[16], respectively. PGSs for all phenotypes assessed were significant predictors of their target trait in G&H (Fig. 3). For prediction in G&H, the incremental $R^2$ for BP was low (-1.8%), but it was higher for lipids and BMI, ranging from 3.9% to 6.7%. Relative accuracy of PGS in G&H versus eMERGE, determined by the ratio of incremental AUC or $R^2$, was close to 1 for HDL-C, TG, SBP, and DBP, and lower for CAD (42%, 95% CI: 30–59%) and BMI (78%, 95% CI: 68–88%; Supplementary Data 5). Amongst the cardiometabolic traits, prediction of LDL-C had the lowest relative accuracy (66%, 95% CI: 53–79%), probably due to the fact that we did not adjust for statin usage since medication data were not available in eMERGE, and British Pakistani and Bangladeshi individuals were more likely to be treated with statins[17]. Incremental $R^2$ for the PGS for LDL-C increased from 3.9% (3.3–4.5%) to 6.2% (5.3–7.1%) when using statin-adjusted LDL-C in G&H (Supplementary Data 5), although the heritability for statin-adjusted and unadjusted LDL-C was not significantly different (Supplementary Data 7; one-sided $p$-value from $z$ test = 0.34).

We explored the factors that may impact the relative accuracy of PGSs. We considered the effect on the relative accuracy of the trans-ancestry genetic correlation, ratio of heritability estimates in G&H versus eMERGE, as well as the product of the previous two terms. However, none of them showed a significant association with the relative PGS performance (Fig. S7).

To assess whether the performance of PGS based on European ancestry GWAS could be improved in British Pakistani and Bangladeshi samples, we next constructed PGS using the clumping and $p$-value thresholding (C + T) method and optimised them separately within G&H and eMERGE using 10-fold cross-validation. The numbers of SNPs in the most frequently selected best C + T PGSs are similar between eMERGE and G&H, and PGSs for lipids contained fewer SNPs (194–454) than other traits (>20,000; Supplementary Data 12, Fig. S8). C + T PGSs and PGSs from the PGS Catalog showed similar performance in G&H across traits, although they were optimised in different ancestry populations (British Pakistani and Bangladeshi and primarily European ancestry, respectively; Fig. S9).

We then assessed whether PGS methods that account for ancestry differences improved predictive accuracy in G&H. PGSs were constructed using a meta-score strategy[18] and using PRS-CSx[19], both integrating the European ancestry GWAS and that from UKBB South Asian ancestry samples. The improvement in accuracy was modest (0.3–10.5%) (Fig. S10). This may be due to the low sample sizes in the UKBB South Asian ancestry GWASs.

## Modest improvement in CAD risk prediction by adding PGS to clinical risk score

A CAD PGS derived from European ancestry GWAS summary statistics and tuned in South Asian ancestry individuals from UKBB[15] (PGS000296 in the PGS Catalog), showed the highest predictive accuracy in British Pakistani and Bangladeshi individuals in G&H. This score had an OR per SD of 1.63 (95% CI: 1.51–1.76) and incremental AUC of 0.009 (95% CI: 0.006–0.012; Supplementary Data 5). Individuals in the top quintile of PGS were predicted to have a 2.2-fold increase (95% CI: 1.78–2.76) in disease risk relative to the middle quintile (quintiles were determined in controls; Fig. S11). We investigated the additional predictive power of PGS on top of established clinical risk factors for CAD, and the net reclassification improvement (NRI) achieved by adding the PGS to a clinical risk score.

To calculate the clinical risk score, we used the QRISK3 algorithm to estimate 10-year risk of cardiovascular disease at a baseline time point, selected so that the participants in G&H had about 10 years of follow-up. QRISK3 was a strong predictor of CAD events and had a concordance index (C-index) of 0.843 (95% CI: 0.828–0.858; Fig. S12, Supplementary Data 13). Consistent with previous findings in European ancestry individuals[8], the CAD PGS was uncorrelated with QRISK3 (Pearson's correlation coefficient $r = -0.0056$ and $p$-value = 0.62). The integrated score combining QRISK3 and the CAD PGS had a non-significant improvement in the C-index (0.853, 95% CI: 0.838–0.867) but a significant improvement in reclassification (categorical NRI: 3.9%; 95% CI: 0.9–7.0%) using a 10-year risk threshold of 10% based on the threshold for preventive intervention with statin treatment recommended by National Institute for Health and Care Excellence[20]. The integrated score reclassified 3.2% of the population as high risk and 2.5% as low risk (Supplementary Data 13). This improvement was mostly driven by the enhanced identification of CAD cases in people between 25 and 54 years of age (NRI in cases being 7.0% vs. NRI in controls being −1.2%), and of controls in people between 55 and 84 of age (NRI in cases being 0.0% vs. NRI in controls being 6.8%) (Fig. 4, Supplementary Data 13). The QRISK3 classified most (91.4%) of the individuals at 55–84 years old as high risk. Using the integrated score, 7.6% of the individuals older than 55 years were down-classified from high to low risk (Supplementary Data 13). Using continuous NRI, the integrated score showed significant improvement (27.0%; 95% CI: 17.7%–36.2%) and similar trends in age groups (Fig. S13, Supplementary Data 13). To assess the potential effects of missingness of QRISK3 variables (Fig. S3), we included additional data for HDL-C and TC that were measured more recently, which were not used in the above standard method (Supplementary Methods). The new QRISK3 score was more accurate with the C-index increased to 0.851, but we still

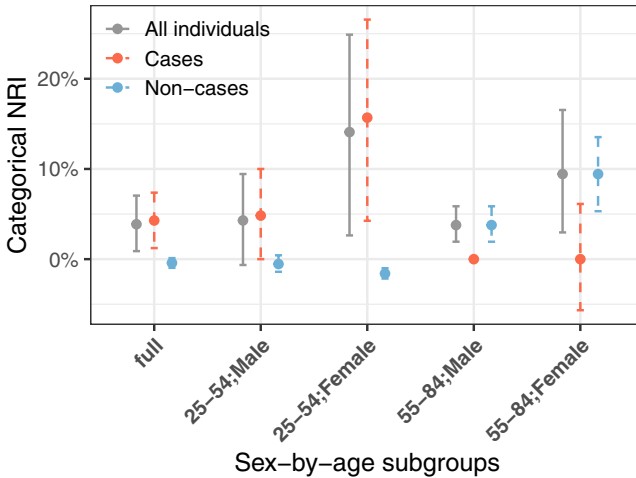

**Fig. 4 | Net reclassification index (NRI) for coronary artery disease with the addition of a polygenic score to QRISK3.** Estimates for categorical NRI for the integrated score compared to QRISK3 in all samples (n = 420 unrelated cases and 7702 unrelated non-cases) as well as in age-by-sex subgroups (n = 207 and 2779 in males aged 25–54; n = 51 and 4187 in females aged 25–54; n = 114 and 344 in males aged 55–84; n = 48 and 392 in females aged 55–84) are shown. Red indicates NRI in cases and blue in controls. The error bars indicate 95% confidence intervals estimated using the bootstrap method.

observed a positive categorical NRI (3.33%; 95% CI: 0.01–6.74) for the integrated score (Supplementary Data 13).

### Estimated causal effects of CAD risk factors largely consistent

We carried out two-sample Mendelian randomisation analyses to estimate potentially causal effects of the risk factors on CAD in G&H and compared findings with European ancestry samples from eMERGE. For G&H, we used transferable loci as genetic instruments to benefit from the precision of largely European ancestry discovery GWAS whilst ensuring only valid instruments are used. In eMERGE, estimates of causal effects for BMI, BP, and lipids, except TG, were statistically significant (Fig. 5). Consistent with this, we found that higher BMI (OR = 1.73, p-value = 0.01), higher LDL-C (OR = 1.55, p-value = $4 \times 10^{-4}$) and lower HDL-C levels (OR = 0.75, p-value = $8 \times 10^{-3}$) were associated with increased risk of CAD in G&H. The effects for SBP and DBP were not statistically significant in G&H. However, both had relatively small numbers of loci as instruments and confidence intervals of the effect estimates were wide.

We also compared different strategies for instrument selection in G&H, such as using all loci associated at genome-wide significance in European ancestry GWAS for the risk factors (Fig. S14). When following the standard approach of using an independent ancestry-matched sample (UKBB South Asian ancestry) to derive the instruments, an insufficient number of genome-wide significant instruments ($p < 5 \times 10^{-8}$) were identified (Fig. S15). To address this, we also tested a less stringent p-value threshold ($p < 5 \times 10^{-5}$) for selecting instruments. For the lipid biomarkers, the results were consistent regardless of which loci were chosen as instruments. However, the association of BMI with CAD was significant only for transferable loci (Fig. S14).

We found evidence of heterogeneity between MR estimates based on Cochran's Q statistic for DBP when using the trait loci from European ancestry GWAS as instruments (p-value = 0.04), LDL-C when using the UKBB South Asian ancestry-ascertained loci (p-value = 0.02) and HDL-C for transferable loci (p-value = $1 \times 10^{-3}$). However, the results of the weighted median and weighted mode models were consistent with those obtained by the inverse-variance weighted Mendelian randomisation model (Supplementary Data 14). We also carried out a multivariable MR for the lipid biomarkers to adjust for potential horizontal pleiotropy. Effect estimates remained highly

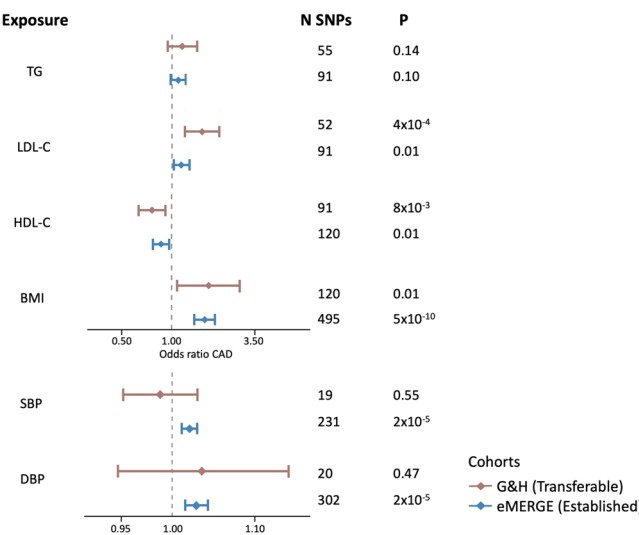

**Fig. 5 | Mendelian randomisation estimates of risk factors on coronary artery disease in European (eMERGE) and British South Asian (G&H) ancestry individuals.** Two-sample Mendelian randomisation (MR) estimates for the causal effects are presented based on genetic instrument variables identified from EUR discovery GWAS for each risk factor. All independent genome-wide significant loci were used as instruments for eMERGE and only the transferable loci for G&H. Effect estimates are presented as odds ratios with 95% confidence intervals per standard deviation increase in the reported unit of the trait: triglycerides (TG), systolic blood pressure (SBP), low-density lipoprotein cholesterol (LDL-C), high-density lipoprotein cholesterol (HDL-C), diastolic blood pressure (DBP), body mass index (BMI). The two-sided p-value (P; not adjusted for multiple comparisons) and the number of single nucleotide polymorphism instruments (N SNPs) included in the MR analysis are shown for each exposure. GWAS for CAD was performed in n = 22,008 (1110 cases) samples from G&H, and n = 32,816 (6815 cases) unrelated samples from eMERGE.

consistent with those observed in the univariable MR (Supplementary Data 15).

## Discussion

We conducted the first study to systematically assess the transferability of genetic loci and PGSs for cardiometabolic traits in individuals of South Asian descent with real-world clinical data, using ~22,000 individuals from the G&H cohort. For lipids and blood pressure, we found evidence that causal genetic variants at published loci are widely shared with European ancestry populations. The prediction accuracy of PGSs derived from European ancestry GWASs for these traits was similar between G&H and European ancestry samples. However, the predictive performance of BMI and CAD PGS was reduced by 22% and 58%, respectively (for the PGS Catalog scores) and CAD also had fewer transferable loci. A CAD PGS optimised for South Asian ancestry individuals nonetheless yielded an appreciable improvement in risk reclassification (categorical NRI = 3.9%; 95% CI: 0.9–7.0%) when combined with the QRISK3 clinical risk score.

Other genetic studies of CAD and related traits that have evaluated reproducibility of loci in South Asian ancestry populations have either been limited by small sample sizes or have restricted their comparisons to the index SNP identified in the GWAS, which does not take LD or statistical power for replication into account[21–23]. A recent study compared genetic determinants of >200 lipid metabolites in 5000 South Asians from Pakistan and 13,000 Europeans and found high overlap in the detected associations[24]. Using a new method, our paper goes further by empirically demonstrating that, in most cases where loci do not replicate, it is due to the lack of power. These findings suggest that, in large part, the genes and pathways that influence the tested cardiometabolic traits are shared between these ancestrally divergent populations. One surprising finding was that the major

LDL-C locus at *CETP* was not associated with this biomarker in G&H but exhibited pleiotropic effects on HDL-C. This is consistent with the observation from a recent study in ancestrally diverse individuals[25]. Abnormalities in *CETP* are linked to accelerated atherosclerosis and might play an important role in increasing risk in South Asian ancestry individuals[26].

For benchmarking we also assessed the transferability of genetic loci in European-ancestry participants from eMERGE. For HDL-C, triglycerides, and blood pressure, the PAT ratio was close to one (Table 1). For LDL-C, we observed a low PAT ratio (0.60, one-sided binomial $p$-value = $1.6 \times 10^{-5}$), probably because statin-adjust data were used in the discovery GWAS but lipid-lowering medication data were not available in eMERGE. For CAD, eMERGE had more cases (6815 vs. 1110), and we were able to replicate 31 loci (9 in G&H). Nevertheless, this is significantly lower than what would be expected given the replication power for eMERGE. In fact, the PAT ratio (PAT = 0.69, one-sided binomial $p$-value = $6 \times 10^{-4}$) was similar to G&H (PAT = 0.62, one-sided binomial $p$-value = 0.05). The lower PAT ratio for CAD in both G&H and eMERGE may indicate that it is a more complex outcome than measurable continuous biomarkers and the definition of CAD is potentially affected by cohort-specific factors such as how the diagnosis is coded in different health care systems. Procedure codes were not available in eMERGE and thus were not considered in defining CAD cases, which adds another source of uncertainty to the comparison.

BMI had the lowest proportion of transferable loci with shared causal variants as well as lower accuracy of the PGS in G&H relative to samples of European descent in eMERGE and a genetic correlation nominally significantly lower than one. South Asian ancestry individuals are known to have higher visceral fat at the same BMI compared to European ancestry individuals in Western countries[27,28]. Consistent with this, the estimated causal effect of BMI on CAD was significant only when using the transferable loci as instruments in the Mendelian randomisation analysis. Visceral adiposity is a strong risk factor for cardiometabolic diseases, independent of total fat mass; these findings warrant further study and may suggest that BMI may not be an optimal biomarker of adiposity in South Asian ancestry individuals[29].

We observed variable performances of PGS developed in European ancestry and applied in British Pakistani and Bangladeshi individuals for the cardiometabolic traits that were investigated in this work, with relative accuracy in G&H versus eMERGE ranging from 131% for DBP to 42% for CAD. Consistent with previous studies[30,31], PGSs for HDL-C and triglycerides had similar predictive accuracy between the two ancestry groups. It is perhaps not surprising to observe a reduction in the accuracy of the CAD PGS in BPBs, given that it did not transfer well even within European-ancestry subpopulations[32]. Medication data were not available in eMERGE, we thus compared the PGS accuracies using the unadjusted, "highest ever" measurements for LDL-C and blood pressure. In G&H where medication data were available, PGSs for SBP and DBP showed similar performance in predicting the adjusted and unadjusted values (Supplementary Data 5). Unlike HDL-C and TG which showed high relative accuracy, PGS for LDL-C showed lower accuracy in G&H than eMERGE, but we were limited in drawing any conclusion because we did not correct for statin usage and PGS showed higher accuracy in G&H when using statin-adjusted LDL-C.

We explored the factors that may impact the relative accuracy of PGSs. Based on a recently proposed theory, relative accuracy is proportional to the product of the trans-ancestry genetic correlation and the ratio of heritability estimates[7]. Neither the trans-ancestry genetic correlation nor the heritability of the trait was associated with the relative PGS performance. This may be because the theory was derived for PGSs based on genome-wide significant SNPs (whereas our PGSs include many SNPs with less significant $p$-values), and because the relative accuracy also depends on differences in allele frequencies and

LD patterns at these SNPs between populations, which we have not factored in and may differ between traits.

Several groups have shown improvements in PGS performance in diverse ancestry groups when incorporating summary statistics from ancestry-matched samples[18,19,33]. Incorporating UKBB South Asian ancestry GWAS data in meta-PGSs proposed by Marquez-Luna et al. [18,33] and using PRS-CSx[19] did not show a large improvement in G&H. A likely reason is the limited sample size of the South Asian ancestry samples in UKBB for some of the traits. Larger samples of South Asian ancestry individuals are needed to examine if ancestry-matched GWAS data can improve prediction accuracy over and above what would be expected from the increased sample size. The increased value of increasing European-ancestry samples versus diversifying ancestries in GWAS will depend on the extent to which the causal variants are shared. For traits for which the causal variants are shared, there is more to be gained from more powerful European ancestry GWASs, even without adding samples of the target ancestry. However, increasing diversity in GWASs will greatly improve the resolution of fine-mapping and the power to identify the causal variants by leveraging the LD differences across ancestries[31,34].

We assessed the clinical value of the PGS for CAD on top of the traditional clinical risk factors captured in the QRISK3 algorithm. Similar work has been done previously in research cohorts[8–10,35]; our study represents an important addition since it captures the noise with which QRISK3 is actually measured within a real-world clinical setting (as opposed to using comprehensive measures taken for research purposes), which may affect the performance of integrated risk models combining these factors with PGSs. We note that only about 4% of the ~8 million individuals used for developing QRISK3 were of South Asian ancestry[36], and the weights for each conventional risk factor might not be optimal for South Asian ancestry individuals. QRISK3 was developed to predict cardiovascular disease (CVD), which is a composite outcome of CAD and stroke. However, our analysis focused on CAD, which is an important component of CVD and the main focus in GWASs and genetic prediction studies. The PGS for CAD developed by Wang et al. showed robust association with CAD in G&H, with a similar OR per SD in PGS (1.63, 95% CI: 1.51–1.76) as in their study (1.60, 95% CI: 1.32–1.94)[15]. The integrated score combining PGS and QRISK3 showed significant reclassification improvement against QRISK3 alone (NRI 3.9% (95% CI: 0.9–7.0%)). Previous studies in UKBB European ancestry samples reported similar improvement, with NRI estimates of 3.5% (95% CI: 2.4–4.5%)[8] and 3.7% (95% CI: 3.0–4.4%)[35] in two different analyses using CAD as the outcome. However, these NRI estimates are probably affected by using UKBB samples that are healthier than the general UK population without recalibrating risk to a primary care setting[9]. In G&H, the PGS improved the identification of high-risk individuals in people younger than 55 years as well as low-risk individuals in people older than 55 years, both of which are important in a clinical setting. We anticipate that, like European ancestry individuals[8,9,35], the British Pakistani and Bangladeshi community (and potentially other South Asian ancestry populations) would also benefit from the use of integrating PGS in primary prevention settings.

Mendelian randomisation has emerged as a powerful tool to explore the potential causal effects of risk factors on disease outcomes. Statistical power can be the limiting factor when extending these analyses to ancestrally diverse populations because independent ancestry-matched GWAS for risk factors of interest may not be sufficiently large. To increase power to estimate the MR causal effects of risk factor traits on CAD in British Pakistanis and Bangladeshis, we used genetic instruments derived from large European ancestry GWAS. Some of the loci may be invalid instruments for other populations. However, restricting the published loci to the ones that were transferable in this population successfully addressed this issue for BMI and shows promise as a new approach for trans-ancestry Mendelian

randomisation. An assumption that requires further study is whether the effect sizes of transferable loci are the same for each ancestry group. Future research should also further investigate the impact of pleiotropy on the causal effect estimate for HDL-C.

Our study has several limitations. Firstly, due to the limited sample size in each age-by-sex subgroup, we could not recalibrate risk prediction models in G&H to what would be expected in an unbiased primary care setting[9]. Secondly, for the comparisons of results in G&H with other cohorts, it should be noted that each of the cohorts examined here is unique. We selected eMERGE, which is also based on electronic health records, for comparison with G&H. Although UK Biobank would have been a better match in terms of country, we were unable to use it for comparison because it was one of the studies included in the published GWAS meta-analyses for the cardiometabolic traits that formed the source of locus discovery as well as the PRS training data. Differences in ascertainment (including the age distribution) and clinical measurements within different cohorts and healthcare systems may have impacted the genetic associations. Different genotyping arrays and imputation panels of different sizes and ancestries were used in G&H and eMERGE, which might lead to potential bias in the comparisons of the two cohorts. The lack of medication data in eMERGE meant we were limited in the conclusions we could draw from comparisons of heritability, transferability (PAT ratio), and PGS performance for LDL-C and blood pressure data. G&H is enriched for young people (median age 40 years old), given that the median age of onset for CAD is 55, some young individuals in this cohort will develop CAD beyond the observation period, which might lead to the underestimation of accuracy and clinical value of the CAD PGS. Finally, our transferability analyses evaluated whether there is evidence for a directionally consistent association in G&H but we did not assess heterogeneity in effect sizes which would also impact genetic risk prediction.

In conclusion, our work provides the first comprehensive assessment of the transferability of cardiometabolic loci to a non-European ancestry population and its impact on two key applications of genetics, causal inference and risk prediction. Our protocol and our new approach for transferability can serve as methodological standards in this developing field. We have shown high transferability of GWAS loci across several cardiometabolic traits between European ancestry and British Pakistani and Bangladeshi populations. The performance of PGSs is trait-specific. Our results suggested there would be clinical value in adding PGS to conventional risk factors in the prediction of CAD in primary care settings to improve the more efficient use of preventive interventions, such as lipid-lowering medications. Our investigation contributes to the increasing representation of individuals of diverse ancestry and varying socio-economic status in research studies, which we hope will help to decrease health disparities.

## Methods

### Genes & Health cohort

Genes & Health (G&H) is a community-based cohort of British Pakistani and Bangladeshi individuals recruited primarily in East London[13]. All participants have consented for lifelong electronic health record access and genetic analysis. The study was approved by the London South East NRES Committee of the Health Research Authority (14/LO/1240). 97.4% of participants in G&H are in the lowest two quintiles of the Index of Multiple Deprivation in the UK. About two-thirds are British Bangladeshi and the remainder British Pakistani. The median age at recruitment was 37 (interquartile range [IQR] = 16) and 43 (IQR = 19) years for female and male participants, respectively (Fig. S1). The cohort is broadly representative of the background population with regard to age, but slightly over-sampled females and those with medical problems since two-thirds of people were recruited in healthcare settings such as GP surgeries[13].

### Quality control and imputation of genotype data from Genes & Health

We used the 2020 February data release which contained 28,022 individuals genotyped on the Illumina Infinium Global Screening Array v3 with additional multi-disease variants. Of these, 22,490 (80%) individuals had linkage to primary or secondary care data, of which 56.5% were female. Quality control of genotype data was performed using Illumina's GenomeStudio and plink v1.9. We removed variants with low call rate (<0.99), rare variants with minor allele frequency (MAF) < 1%, and variants that failed the Hardy–Weinberg test ($p < 1 \times 10^{-6}$) in a subset of samples with low level of autozygosity (Supplementary Methods). We excluded individuals who did not have Bangladeshi or Pakistani ancestry (further than +/− 3 standard deviations [SD] from the mean of PC1 for the individuals who self-reported as coming from that group), and those who self-reported as coming from other ethnic groups or who did not report this information (Fig. S2).

We used the Michigan Imputation Server[37] to perform imputation with the GenomeAsia pilot reference panel[38], imputing from 336,133 autosomal, biallelic SNPs with matched alleles. Eagle v2.4 and Minimac v4 were used for phasing and imputation, respectively. We excluded SNPs with imputation INFO score <0.3 or MAF < 0.1%, which left 9,527,863 autosomal SNPs.

We applied more stringent QC on GWAS results using the EasyQC package followed[39]: allele mismatch and allele frequency difference of >0.2 with reference panel, imputation INFO score <0.7 (<0.9 for downstream analysis i.e. correlation and colocalisation), MAF < 0.5% (<1% for downstream analysis i.e. correlation and colocalisation).

### Quality control and imputation of genotype data from eMERGE

We used European ancestry samples from the eMERGE cohort (henceforth eMERGE), a consortium of US medical research institutions, to carry out comparisons with G&H. Network Phase III data ($N = 61,377$) were downloaded from dbGaP (Accession number: phs001584.v1.p1). Quality control of genotype data and imputation to the Human Reference Consortium (HRC) reference panel have been described previously[40]. We projected eMERGE participants onto the PC space generated from the 1000 Genomes project phase 3 dataset and applied Uniform Manifold Approximation and Projection (UMAP)[41], and identified 43,877 European ancestry individuals. Well-imputed (INFO ≥ 0.3) bi-allelic SNPs with MAF ≥ 0.1% ($N = 11,625,805$) were retained for downstream analysis.

### Phenotype and covariate definitions from electronic health-record data

Coronary artery disease (CAD) cases and controls in G&H were defined using the same ICD10 and OPCS4 codes as Khera et al. [42] (Supplementary Data 1; Supplementary Methods). Data processing for BMI, lipids, and blood pressure is in Supplementary Methods. Both the highest and medication-adjusted measurements were available in G&H. Sample sizes are shown in Table 1 (all individuals) and Supplementary Data 2 (unrelated).

We calculated the QRISK3 10-year predicted risk for CAD[36] in G&H using the R package "QRISK3" v0.3.0[43]. We used data available up until 1 January 2010 to calculate QRISK3 (Supplementary Methods). Definitions of variables in the QRISK3 algorithm are shown in Supplementary Data 3, following[8].

Phenotype data in eMERGE were downloaded from dbGaP (phs001584.v1.p1, phs000888.v1.p1, and phs001584.v2.p2; Supplementary Methods). Data on medications affecting lipid and BP measurements were not available, so the highest measurements for LDL, TC, SBP, and DBP were used when comparing heritability estimates and performance of PGSs with G&H in order to minimise the effects of medications.

## Genome-wide association analyses in Genes & Health

GWAS was performed with SAIGE[44] and adjusted for age, age$^2$, sex and the first twenty principal components. For total cholesterol and LDL-C, adjustments were made for use of statins as described above.

## Heritability and trans-ancestry genetic correlations

Datasets that were used in analyses are provided in Supplementary Data 4. We used GCTA to estimate SNP heritability in unrelated individuals from G&H and eMERGE, correcting for age, sex, and first 10 genetic PCs[45]. For CAD, we estimated SNP heritability on the liability scale using 6.7% as the prevalence estimate in the US[46], and 3.33% for the UK background population from which G&H is sampled, defined as all people from South Asian ethnicities ($N = 255,066$ aged ≥20 years) registered with a primary health physician/GP in four east London boroughs.

For the genetic correlation analyses, we used GWAS summary statistics generated in European ancestry individuals from UK Biobank (UKBB), since we needed a larger sample size of ancestrally homogeneous individuals than is available through eMERGE to obtain accurate estimates. We used Popcorn (https://github.com/brielin/Popcorn) to estimate the trans-ancestry genetic correlations between G&H and UKBB European ancestry individuals while accounting for differences in LD structure (Supplementary Methods)[47]. A two-sided $p$-value < 0.05 indicated that the genetic correlation was significantly different from one.

## Assessment of transferability of trait loci

Previous studies that evaluated the reproducibility of GWAS loci in South Asian individuals did not formally account for differences in power or LD patterns[21–23]. We assessed whether published trait-associated loci were reproducible in G&H (Supplementary Data 4), i.e. whether a locus affects the same trait in both populations, regardless of effect sizes. Credible sets for trait loci were generated and consisted of lead (independent) variant plus proxy SNPs ($r^2 \geq 0.8$) within a 50 kb window (based on the European ancestry 1000 Genomes data) of the sentinel variant and with $p$-value $< 100 \times p_{sentinel}$. The locus was defined as being 'transferable' if at least one variant from the credible set was associated at two-sided $p < 0.05$ with the relevant trait in G&H, and the direction of effect matched in both datasets. For loci harbouring multiple signals, we only kept the most strongly associated variant (i.e. smallest $p$-value). The statistical power to observe an association of a given locus in G&H was calculated using alpha = 0.05, the effect size estimate for the lead variant from the European ancestry discovery GWAS, and the allele frequency of the variant and sample size in G&H (Supplementary Methods). For SBP and DBP, the raw measurements were used in the discovery GWAS, we thus calculated power with effect size estimates in UK Biobank European-ancestry individuals by Neale's group where normalised blood pressure values were used. The power estimates were summed up across published loci for a given trait to give an estimate of the number of loci expected to be significantly associated in G&H. This is the expected number if all loci are transferable and account for the statistical power for replication. We calculated the power-adjusted transferability (PAT) ratio by dividing the observed number of loci with $p < 0.05$ amongst the published loci in G&H over the expected number. To our knowledge, this is a novel approach for assessing the reproducibility of GWAS findings.

We also highlighted published trait loci that we deemed to be 'non-transferable' despite sufficient statistical power: they contained at least one variant in the credible set with >80% power for replication and yet none of the variants in the credible set had $p < 0.05$ and no variant within 50 kb of locus had $p < 1 \times 10^{-3}$ in G&H. LocusZoom (http://locuszoom.org/) was used to create regional association plots.

## Trans-ancestry colocalisation

We used the Trans-ancestry colocalisation method (TAColoc) (https://github.com/KarolineKuchenbaecker/TEColoc)[30] which tests whether a specific locus has the same causal variant in two groups with different ancestry, and applied it to G&H and UKBB European ancestry individuals (Supplementary Methods).

## Construction of polygenic scores

We evaluated the performance of PGSs in G&H and eMERGE. We first assessed PGSs that were previously constructed (mostly optimised in European ancestry samples) from the PGS Catalog[48]. We restricted to 7,353,388 bi-allelic SNPs that had INFO ≥ 0.3 and MAF ≥ 0.1% in both eMERGE and G&H. Variant information in existing PGS was harmonised to GRCh37 using dbSNP mappings from Ensembl Variation and liftover. We calculated PGSs as weighted sums of imputed allele dosages using plink2.0–score function. When multiple PGSs were available in the PGS Catalog, we reported the best score per trait. The details of the scores are in Supplementary Data 5.

We also calculated PGSs using the clumping and $p$-value thresholding method (C + T) and optimised PGSs in G&H and eMERGE separately using 10-fold cross-validation (Supplementary Methods). Lastly, in G&H we calculated meta-PGSs proposed by Marquez-Luna et al. [18] and PGS using the PRS-CSx method[19] that incorporated GWAS summary data from the panUKBB South Asian-ancestry individuals (Supplementary Methods).

## Assessment of PGS accuracy and clinical performance

Age at recruitment was used as a covariate for analysis of disease status, and age at measurement for analysis of quantitative traits. PGSs were standardised to a mean of 0 and SD of 1. We fitted the following two models: (1) the full model which had PGS and covariates namely sex, age, age$^2$, and the first 10 genetic PCs, and (2) the reference model which accounted for the covariates only. For continuous risk factors, linear regression was fitted, and the gain in $R^2$ when adding PGS as an additional predictor, or incremental $R^2$, was calculated as the difference between the $R^2$ of the full model and the reference model. Logistic regression was used to assess the associations between PGSs and CAD. The area under the receiver operating characteristic curve (AUC) was estimated for both models with the R package "pROC" v1.16.2 and incremental AUC was calculated similarly. We performed bootstrap resampling of individuals 1000 times to estimate the 95% confidence intervals for incremental $R^2$ and incremental AUC. We estimated the effect size per SD of PGS from the full model. Effect size for quintiles, and for the top 10% versus middle 40–60% was reported as well. Relative accuracy was calculated as the ratio of incremental AUC (or incremental $R^2$ for continuous traits) in G&H to that in eMERGE.

QRISK3 scores were calculated for 8112 unrelated individuals as described in Supplementary Methods (420 CAD cases and 7702 controls). We followed Riveros-Mckay et al. [8] to integrate QRISK3 scores with the PGS for CAD developed by Wang et al. [15]. Cox regression was performed using the R package "survival" v3.2-7. The concordance indices (C-indices) of the following models were compared: (1) age at assessment + sex, (2) PGS + age at assessment + sex, (3) QRISK3, and (4) the integrated score. We calculated the continuous net reclassification index (NRI) and categorical NRI (using 10% as the threshold to classify high-risk individuals) for the integrated score compared to QRISK3 alone. NRI was calculated as the sum of NRI for cases and NRI for controls (noncases):

$$\text{NRI} = P(\text{up}|\text{case}) - P(\text{down}|\text{case}) + P(\text{down}|\text{noncase}) - P(\text{up}|\text{noncase})$$

For continuous NRI, $P(\text{up}|\text{case})$ and $P(\text{down}|\text{case})$ indicate the proportions of cases that had higher or lower risk estimates using the integrated score, respectively. For categorical NRI, $P(\text{up}|\text{case})$ indicates the proportions of cases that were reclassified as high-risk individuals

(i.e. with <10% risk by QRISK3 but >10% by the integrated scores). We calculated NRI in two age groups (25–54 versus 55–84 years old at baseline, chosen since the average age of onset in this cohort was 55.3 years old), as well as in age-by-sex subgroups. Bootstrap resampling (1000 times) was used to estimate confidence intervals for NRI. All reported $p$-values are two-sided.

### Mendelian randomisation analysis

We modelled liability to CAD as our outcome within a univariable two-sample Mendelian randomisation (MR)[49] framework using the cardio-metabolic traits (BMI, SBP, DBP, LDL-C, HDL-C, TG) as exposures. To identify genetic instruments for the exposure, we explored three alternative approaches: (a) published loci significant at $p < 5 \times 10^{-8}$ in the original European ancestry GWAS; (b) transferable loci defined as described in the transferability section of the methods, taking the effect size from the original European ancestry GWAS; and (c) loci significant at $p < 5 \times 10^{-8}$ in the South Asian ancestry group of the Pan-UKBB GWAS, LD-clumped to an $r^2 < 0.2$ with a LD window of 50 kb, based on South Asian 1000 Genomes project LD reference. Where insufficient numbers of genome-wide significant instruments were identified, we used a more permissive $p$-value threshold of $p < 5 \times 10^{-5}$ for instrument selection in UKBB South Asian. The primary Mendelian randomisation analysis was performed using, as outcome, summary association data from the G&H CAD GWAS performed as described above, using the inverse-variance weighted method under a random effect model, implemented with the TwoSampleMR R package v0.5.5[50]. For comparison, a two-sample Mendelian randomisation approach was also performed using summary data for CAD from eMERGE and established loci significant at $p < 5 \times 10^{-8}$ in the original European ancestry GWAS. We also undertook several sensitivity analyses. In brief, we evaluated the Egger intercept to assess directional pleiotropy and Cochran's $Q$ statistic[51] as an indicator of heterogeneity. Mendelian randomisation analysis using MR pleiotropy residual sum and outliers methods (MR-PRESSO)[52], weighted median[53] and weighted mode methods[54] models were additionally performed in the presence of heterogeneity. To investigate the individual direct effect of HDL-C, LDL-C, and TG on the risk of CAD and simultaneously account for horizontal pleiotropy, we replicated the analysis in a multivariable MR (MVMR)[55] setting with the TwoSampleMR R package. Genetic instruments for the HDL-C, LDL-C, and TG joint exposure were selected from samples with European ancestry in UKBB, if associated with at least one of the three lipids ($p < 5 \times 10^{-8}$). The instruments were filtered based on MAF > 0.005 and LD-clumped to an $r^2 < 0.01$ with a window of 50 kb, based on the EUR 1000 Genome project LD reference, using plink2. All reported $p$-values are two-sided.

### Reporting summary

Further information on research design is available in the Nature Research Reporting Summary linked to this article.

## Data availability

Genes & Heath imputed genotype data (GRCh 37) have been deposited in EGA under study accession number: EGAS00001005373 (https://ega-archive.org/datasets/EGAD00001007815). The electronic health records from Genes & Health are available under restricted access for bona fide research; researchers wishing to access them should apply to the G&H Executive (www.genesandhealth.org/research/scientists-using-genes-health-scientific-research). GWAS summary statistics generated in Genes & Health are available at www.genesandhealth.org/research/scientific-data-downloads. The transferable loci generated in this study are provided in the Supplementary Data file. Publicly available GWAS summary statistics that were used in this study (Supplementary Data 4) are available via the CARDIoGRAMplusC4D Consortium (http://www.cardiogramplusc4d.org), GIANT (https://portals.broadinstitute.org/collaboration/giant/index.php/Main_Page),

GLGC (http://csg.sph.umich.edu/willer/public/lipids2017/), and GWAS Atlas (https://atlas.ctglab.nl/traitDB/). SNPs and the weights for polygenic risk scores are available in the PGS Catalog (www.pgscatalog.org) and score IDs are provided in Supplementary Data 5.

## Code availability

Code used to assess the transferability of genetic loci and calculate the PAT ratio is available at https://github.com/Nsallah1/GH_Manuscript. TAColoc used to perform trans-ancestry colocalisation analysis is available at https://github.com/KarolineKuchenbaecker/TEColoc. Code used to assess the accuracy and clinical utility of polygenic risk scores is available at https://github.com/QinqinHuang/GnH28k_polygenic_scores[56]. All additional software is available online.

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

## Acknowledgements

We thank Social Action for Health, Centre of The Cell, members of our
Community Advisory Group, and staff who have recruited and collected

data from volunteers. We thank the NIHR National Biosample Centre (UK Biocentre), the Social Genetic & Developmental Psychiatry Centre (King's College London), Wellcome Sanger Institute, and Broad Institute for sample processing, genotyping, sequencing and variant annotation. We thank Barts Health NHS Trust, NHS Clinical Commissioning Groups (Hackney, Waltham Forest, Tower Hamlets, Newham), East London NHS Foundation Trust, Bradford Teaching Hospitals NHS Foundation Trust, and Public Health England (especially David Wyllie) for GDPR-compliant data sharing. We thank all members from the Genes & Health Research Team (full list in the Supplementary Information). We also thank Sally Hull and Martin Sharp from the primary care data team at QMUL for their help in estimating population prevalence of CAD. Most of all we thank all of the volunteers participating in Genes & Health. Genes & Health is/has recently been core-funded by Wellcome (WT102627, WT210561), the Medical Research Council (UK) (M009017), Higher Education Funding Council for England Catalyst, Barts Charity (845/1796), Health Data Research UK (for London substantive site), and research delivery support from the NHS National Institute for Health Research Clinical Research Network (North Thames). This research was funded in part by the Wellcome Trust Grant 206194 to the Wellcome Sanger Institute. This research was funded in part by the European Research Council (ERC) under the European Union's Horizon 2020 research and innovation programme (Grant agreement No. 948561). This work was supported by core funding from the: British Heart Foundation (RG/13/13/30194; RG/18/13/33946), BHF Cambridge Centre of Research Excellence (RE/13/6/30180), and NIHR Cambridge Biomedical Research Centre (BRC-1215-20014) [*The views expressed are those of the author(s) and not necessarily those of the NIHR or the Department of Health and Social Care]. This work was also supported by Health Data Research UK, which is funded by the UK Medical Research Council, Engineering and Physical Sciences Research Council, Economic and Social Research Council, Department of Health and Social Care (England), Chief Scientist Office of the Scottish Government Health and Social Care Directorates, Health and Social Care Research and Development Division (Welsh Government), Public Health Agency (Northern Ireland), British Heart Foundation and Wellcome. S.A.L. is supported by a Canadian Institutes of Health Research postdoctoral fellowship (MFE-171279). C.G. is supported by the National Institute for Health Research ARC North Thames. M.I. is supported by the Munz Chair of Cardiovascular Prediction and Prevention and the NIHR Cambridge Biomedical Research Centre (BRC-1215-20014). M.I. was also supported by the UK Economic and Social Research Council (ES/T013192/1). R.T.L. and N.S. are supported by the BigData@Heart Consortium funded by the Innovative Medicines Initiative-2 Joint Undertaking under grant agreement No. 116074. R.T.L. is additionally supported by the UCL British Heart Foundation Research Accelerator and has received support from a UK Research and Innovation Rutherford Fellowship hosted by Health Data Research UK (MR/S003754/1). For the purpose of Open Access, the author has applied a CC BY public copyright licence to any Author Accepted Manuscript version arising from this submission.

## Author contributions

K.K., H.C.M., and R.T.L. designed the study and supervised the work. J.W., C.G., R.C.T, H.M., H.C.M., S.F., and D.A.v.H lead and conduct the Genes & Health programme. B.T., S.F., and D.A.v.H extracted and curated the phenotypic data. K.A.H., Q.Q.H., E.A., and D.A.v.H performed genotyping, imputation, and quality control. N.S. and K.K. assessed the transferability of genetic loci. D.D. performed the Mendelian randomisation analysis. Q.Q.H. conducted the polygenic score analysis. S.A.L. provided the curated polygenic scores and helped with scoring. S.H. helped with preparing QRISK3 clinical variables. H.H. helped supervise the work. M.I. helped estimate the accuracy of polygenic scores. All authors were involved in the interpretation of results. Q.Q.H., N.S., D.D., R.T.L., H.C.M., and K.K. drafted the manuscript and all authors read, edited, and approved the final version of the manuscript.

## Competing interests

N.S. is now employed by GlaxoSmithKline. All other authors declare no competing interests.
