## [Peer Review File · Nature Communications]

Transferability of genetic loci and polygenic scores for cardiometabolic traits in British Pakistani and Bangladeshi individualsEditorial Note: This manuscript has been previously reviewed at another journal that is not operating a transparent peer review scheme. This document only contains reviewer comments and rebuttal letters for versions considered at *Nature Communications*.

REVIEWERS' COMMENTS

Reviewer #2 (Remarks to the Author):

The authors have comprehensively addressed my original concerns. This will be an important contribution. I only have a couple minor points for consideration.

Minor:

1. In the Discussion, the authors note "In G&H, the PGS improved identification of high-risk individuals in people younger than 55 years, and correctly down-classified low-risk individuals in people older than 55 years, both of which are important in a clinical setting." I would be more circumspect with the latter phrasing – something like "improved down classification of..."
2. The authors nicely described their rationale for using eMERGE (European ancestry in the US) to compare with Genes & Health (South Asian ancestry in the UK). UKBiobank cannot be used since it is often used in training. And another British cohort from a similar healthcare system was not available. These elements should be incorporated into the limitations section of the Discussion as they may explain some of the observed differences in the Results beyond ancestry.

Reviewer #4 (Remarks to the Author):

I was not an original reviewer, but I have been asked to review this revision, specifically focusing on the authors' response to Reviewer #3. Overall, I found this manuscript very interesting, and I thank the authors and editors for the opportunity to review it. I find the PAT analysis of particular interest. In regards to Reviewer 3's comments, I think the authors have done a good job at addressing each comment to the extent that they are able. I have no further major comments and only a few minor comments for consideration.

MAJOR
None

MINOR

1. I completely agree with Reviewer 3 that calculating PAT in eMERGE cohort provides a valuable comparison, and the results provided in this revision are quite interesting. However, the motivation and interpretation of this analysis (while quite clear in the reviewer/rebuttal document) is not so clear in the main text. It would be helpful to the reader if the authors provide additional motivation and interpretation beyond the statement: "In eMERGE, we observed a similarly low PAT ratio for CAD (0.69, binomial p-value = 6×10^{-4}) (Table 1 and Table S9)."
2. I found it hard to understand the details of the cross validation method used for choosing and validating a C+T PGS. In order to fully understand the process and what was being shown in Supplemental Figure 9, I had to piece together information provided in the rebuttal document, the main text, the supplemental methods, Supplemental Figure 9, and Supplemental Table 12. It would be helpful if the authors could consolidate this information in the supplemental methods and provide a more detailed description of the method and what is ultimately reported.
3. Table 1 has two different asterisk descriptions, and it is unclear what is referring to what.

4. Page 9, line 18 of the main text refers to Figure 9C, but I do not think there is a Figure 9C.

REVIEWERS' COMMENTS

Reviewer #2 (Remarks to the Author):

The authors have comprehensively addressed my original concerns. This will be an important contribution. I only have a couple minor points for consideration.

Minor:

1. In the Discussion, the authors note “In G&H, the PGS improved identification of high-risk individuals in people younger than 55 years, and correctly down-classified low-risk individuals in people older than 55 years, both of which are important in a clinical setting.” I would be more circumspect with the latter phrasing – something like “improved down classification of...”

We thank the reviewer for the suggestion. We now have updated the Discussion: “In G&H, the PGS improved identification of high-risk individuals in people younger than 55 years as well as low-risk individuals in people older than 55 years, both of which are important in a clinical setting”.

2. The authors nicely described their rationale for using eMERGE (European ancestry in the US) to compare with Genes & Health (South Asian ancestry in the UK). UKBiobank cannot be used since it is often used in training. And another British cohort from a similar healthcare system was not available. These elements should be incorporated into the limitations section of the Discussion as they may explain some of the observed differences in the Results beyond ancestry.

We thank the reviewer for the suggestion. We have now included the following sentences in our discussion about the limitations of using eMEREG for the comparison with G&H and cohort-specific ascertainment bias: “We selected eMERGE, which is also based on electronic health records, for comparison with G&H. Although UK Biobank would have been a better match in terms of country, we were unable to use it for comparison because it was one of the studies included in the published GWAS meta-analyses for the cardiometabolic traits that formed the source of locus discovery as well as the PRS training data.”

Reviewer #4 (Remarks to the Author):

I was not an original reviewer, but I have been asked to review this revision, specifically focusing on the authors' response to Reviewer #3. Overall, I found this manuscript very interesting, and I thank the authors and editors for the opportunity to review it. I find the PAT analysis of particular interest. In regards to Reviewer 3's comments, I think the authors have done a good job at addressing each comment to the extent that they are able. I have no further major comments and only a few minor comments for consideration.

We thank the reviewer for contributing to the review of the manuscript and the kind comments.

MAJOR

None

MINOR

1. I completely agree with Reviewer 3 that calculating PAT in eMERGE cohort provides a valuable comparison, and the results provided in this revision are quite interesting. However, the motivation and interpretation of this analysis (while quite clear in the reviewer/rebuttal document) is not so clear in the main text. It would be helpful to the reader if the authors provide additional motivation and interpretation beyond the statement: “In eMERGE, we observed a similarly low PAT ratio for CAD (0.69, binomial p-value = 6×10^{-4}) (Table 1 and Table S9).”

We thank the reviewer for the comment. We have now added a sentence before this statement in the Results on page 5 to explain the motivation, and a paragraph on the interpretation of the observations in the Discussion on page 12.

2. I found it hard to understand the details of the cross validation method used for choosing and validating a C+T PGS. In order to fully understand the process and what was being shown in Supplemental Figure 9, I had to piece together information provided in the rebuttal document, the main text, the supplemental methods, Supplemental Figure 9, and Supplemental Table 12. It would be helpful if the authors could consolidate this information in the supplemental methods and provide a more detailed description of the method and what is ultimately reported.

We thank the reviewer for the suggestion. We have now expanded the “Comparison of performances of polygenic scores between ancestries” section in the Supplementary Methods (page 8). We have given an overview in the first paragraph:

“In this study, we compared the performances of PGSs in BPB people from the G&H cohort and EUR people from the eMERGE cohort. We calculated the PGSs in the two cohorts using the two approaches described below. We reported the results using the PGS Catalog scores in the main text, because using previously published scores developed in external cohorts is less likely to have the issue of overfitting. We also compared the two approaches in Figure S9.”

We then explained the two approaches to calculating the PGSs in detail. For the C+T PGS, we have also added the motivations for using the GWAS data in Table S12 (now Supplementary Data 12) and for cross-validation.

We have also added a summary of the two approaches in the legend for Figure S9 on page 20 and have referred it to Supplementary Methods for detailed methods.

3. Table 1 has two different asterisk descriptions, and it is unclear what is referring to what.

We agree with the reviewer that asterisks in Table 1 are not very clear. We have now moved most information in the table legend or the Methods section, and only kept the asterisks to indicate the slightly different phenotypic definitions used in eMERGE.

4. Page 9, line 18 of the main text refers to Figure 9C, but I do not think there is a Figure 9C.

We thank the reviewer for pointing out the typo. We previously had a supplementary figure for it but removed it later. We have updated the text and it's now referring to Table S5 (now Supplementary Data 5).